# Learning data-efficient coarse-grained molecular dynamics from forces and noise

Aleksander E. P. Durumeric[1,6], Yaoyi Chen [1,6], Aldo S. Pasos-Trejo[2,6], Frank Noé [1,2,3,4] ✉ & Cecilia Clementi [4,5] ✉

Molecular dynamics (MD) simulations are essential for elucidating biomolecular function, yet the computational cost of all-atom models often limits their reach. Machine-learned coarse-grained (MLCG) models offer a solution by simplifying the representation while maintaining near-atomistic accuracy. However, the training of MLCG models currently requires vast amounts of force-labeled sample conformations from reference atomistic MD. Here, we overcome this limitation by unifying the training of MLCG models with the principles of generative diffusion models. We demonstrate that accurate high-dimensional distributions of molecular ensembles can be recovered by integrating traditional force-matching with denoising objectives. This framework enables the construction of physically consistent and stable force fields while reducing atomistic data requirements by up to two orders of magnitude. Validated across diverse protein folds and scales, our work establishes a bridge between molecular dynamics simulation and modern generative learning, substantially lowering the computational cost of constructing accurate MLCG models and broadening their applicability to large biomolecular systems.

Atomistic molecular dynamics (MD) can, in principle, connect the microscopic motion of a large number of atoms to emergent properties[1–4]. However, despite decades of development, its application to large biomolecular systems is hampered by its computational cost. As atomistic MD computes the interactions of all biomolecular and solvent atoms and advances simulation time by a few femtoseconds per step, it is often computationally prohibitive to simulate timescales relevant for biomolecular dynamics. This has led to the creation of coarse-grained (CG) MD models, which use far fewer effective atoms and can employ effectively longer simulation timesteps, resulting in large speedups[5–11]. For example, while atomistic MD could model a solvated protein by explicitly representing each atom in the protein and surrounding water, CG MD could do so by considering only the atoms in the protein backbone[6].

Unfortunately, while atomistic MD has achieved high accuracy[12,13], CG MD has struggled to do so[9–11]. MD propagates each particle in time

according to the forces on each atom—the negative derivatives of a potential energy function, which is generally called a force-field. Especially for biomolecular systems, atomistic force fields have been improved over decades using a combination of quantum mechanical calculations and experimental data. As a result, despite limitations in their functional form, modern atomistic force fields have reached a level of high predictiveness in terms of metastable structures and free energy differences, if sufficient sampling is used[13–18]. However, CG force fields are still far away from their atomistic counterparts in generalization and accuracy. Recent progress has suggested that machine learning (ML) may be central to achieving a comparable level of performance[19–23], as ML can learn crucial many-body interactions which are often missing from traditional CG force-fields[24,25].

Many methods exist to represent and train CG force-fields[8–10]. Training methods typically pursue either a top–down approach, in which the values of macroscopic quantities are matched[10,11], or a

[1]Department of Mathematics and Computer Science, Freie Universität Berlin, Berlin, Germany. [2]Department of Physics, Freie Universität Berlin, Berlin, Germany. [3]AI4Science, Microsoft Research, Berlin, Germany. [4]Department of Chemistry, Rice University, Houston, TX, USA. [5]Center for Theoretical Biological Physics, Rice University, Houston, TX, USA. [6]These authors contributed equally: Aleksander E. P. Durumeric, Yaoyi Chen, Aldo S. Pasos-Trejo. ✉e-mail: frank.noe@fu-berlin.de; cecilia.clementi@fu-berlin.de

bottom-up approach that aims to reproduce the detailed behavior present in a corresponding atomistic model[8,9]. Here we focus on the bottom-up approach following multiscale coarse-graining (also known as "variational force-matching")[26–30]. In terms of the CG force-field representation, it is nontrivial how to design the functional form of the force-field itself to be sufficiently expressive yet practical for a given application. Recent attempts have borrowed function representations from ML[21] (e.g., neural networks[31]), an approach that has seen preliminary success in modeling biomolecules[19,20,22–24,32–39]. However, learning the corresponding force-field has required either substantial computation or a significant amount of data[21]. The resources necessary for the creation of machine-learned coarse-grained (MLCG) force fields represent a significant barrier to the development of the next generation of CG models, which could rival the accuracy of atomistic MD.

Here, we present a strategy for training bottom-up neural-network CG force fields using reference atomistic MD simulation data that is computationally efficient and reduces data requirements by a factor of 100. This is made possible by unifying denoising score matching[40–42] (the key technology underlying denoising diffusion models (DDMs)[43–45], a popular type of ML generative model) with the learning of MLCG force-field parameters based on the "force-matching" approach[28]. This connection, in turn, suggests novel modifications for both classes of models. The corresponding methodologies are implemented in public code bases and demonstrated by creating MLCG force-fields of the Chignolin[46], Trp-Cage[47], NTL9[48], and Ubiquitin proteins using minimal amounts of training data.

## Results
### Combined force matching and denoising for learning CG force-fields

**Learning from forces.** We train a CG neural network potential $U_\theta(\mathbf{R})$ with neural network parameters $\theta$ as a function of the CG coordinates, $\mathbf{R}$. Force-matching minimizes the mean squared difference between the atomistic forces $f(\mathbf{r})$ at configuration $\mathbf{r}$, projected onto the CG space, and the forces of the CG force-field. The CG coordinates $\mathbf{R}$ are usually defined as a linear mapping of the atomistic coordinates: $\mathbf{R} = \mathbf{Mr}$. The effective CG force-field which produces the same equilibrium distribution as the corresponding atomistic system, referred to as the "many body potential of mean force" ($U_{\mathrm{PMF}}$), minimizes the loss function[6,28]:

$$\mathcal{L}(\theta) = \left\langle \| F_\theta(\mathbf{R}) - \mathbf{T}f(\mathbf{r}) \|_2^2 \right\rangle_{\mathbf{r}} \qquad (1)$$

where $F_\theta(\mathbf{R}) = -\nabla U_\theta(\mathbf{R})$ is the CG force-field generated by the learned CG potential, $\langle\cdot\rangle_{\mathbf{r}}$ denotes an ensemble average over the equilibrium distribution of the atomistic system, and $\mathbf{T}$ transforms atomistic forces into forces on the CG particles[28,36,49].

To accurately approximate the potential of mean force $U_{\mathrm{PMF}}$ implied by the atomistic simulation, a flexible functional form is needed for $U_\theta(\mathbf{R})$, and neural networks have been shown to be a formidable choice[21]. However, training neural network CG force-fields for biomolecular systems requires a lot of data, e.g., many diverse configurations with force labels[36]. Here, we propose a much more data-efficient approach that leverages not only the force labels but also the distribution of configurations $\mathbf{R}$ in the training data while avoiding repeated long simulation of the CG model[8,9,33,38,50,51].

**Learning from noise.** The proposed training modifications extend denoising score matching[42], which is the central idea in DDMs[43–45], a class of machine-learning algorithms trained to transform noise variates to approximate the distribution of training data. Analogous to distribution-based force-field learning[8,9], DDMs learn the distribution present in a data sample without pre-recorded force information; instead, this information is learned through the addition and removal of noise.

During DDM training, data are first corrupted to different extents with additive noise, with maximal corruption giving an uninformative prior distribution and no corruption corresponding to the original data distribution. For simplicity in derivation and implementation, the noise is usually set to follow an isotropic Gaussian distribution, and the extent of corruption (i.e., the noise level) corresponds to the variance of the Gaussian noise. This corruption process is captured by introducing a conditional density $\kappa(\mathbf{r}'|\mathbf{r})$, which describes noising a training data sample $\mathbf{r}$ to obtain the noised data $\mathbf{r}'$. We can interpret $-\log \kappa(\mathbf{r}'|\mathbf{r})$ as a "noise energy" and its negative gradients as "noise forces", $F_\kappa = \nabla_{\mathbf{r}'} \log \kappa(\mathbf{r}'|\mathbf{r})$. A neural network $S_\theta(\mathbf{r}')$, called the "score network", is trained to predict the noise forces by minimizing a loss function. This loss function is defined as a sum of individual denoising score matching losses for each distortion level. The loss for a single distortion level can be interpreted as force matching (Eq. 1) further averaged over the noise variables:

$$\mathcal{L}_{\mathrm{DSM}}(\theta) = \left\langle \mathbb{E}_\kappa \| S_\theta(\mathbf{r}') - F_\kappa(\mathbf{r}', \mathbf{r}) \|_2^2 \right\rangle_{\mathbf{r}}, \qquad (2)$$

where the brackets $\langle\cdot\rangle_{\mathbf{r}}$ denote the expectation over the data distribution via $\mathbf{r}$ and $\mathbb{E}_\kappa$ is the expectation over the noise variables $\mathbf{r}'$. DDMs have been shown to be successful in various tasks[52], including modeling CG protein conformational landscapes by fitting the training data distribution[39,53].

**Combining noise and forces in coarse-graining.** We propose to augment Eq. (1) by adapting the denoising described in Eq. (2) for a single noise level through the definition of a modified loss function:

$$\mathcal{L}^\kappa(\theta; \mathbf{T}) = \left\langle \mathbb{E}_\kappa \| F_\theta(\mathbf{R}) - \mathbf{T}f(\mathbf{R}, \mathbf{r}) \|_2^2 \right\rangle_{\mathbf{r}} \qquad (3)$$

where the force $f(\mathbf{R}, \mathbf{r})$ is now defined as a sum of the atomistic MD forces on fine-grained coordinates $\mathbf{r}$ and noise forces defined over an extended system with coordinates $(\mathbf{R}, \mathbf{r})$ with energies specified in thermal units ($k_B T = 1$):

$$f(\mathbf{R}, \mathbf{r}) = \underbrace{-\nabla u(\mathbf{r})}_{\text{MD Forces}} + \underbrace{\nabla \log \kappa(\mathbf{R}, \mathbf{r})}_{\text{Noise Contributions}} . \qquad (4)$$

The force map $\mathbf{T}$ in Eq. (3) projects forces on the extended system to CG coordinates $\mathbf{R}$.

The noise $\kappa(\mathbf{R}, \mathbf{r})$ is utilized as a probabilistic replacement for the CG map $\mathbf{M}$: as where $\mathbf{M}$ deterministically defines the CG positions $\mathbf{R}$ given the fine-grained positions $\mathbf{r}$, $\kappa$ instead describes the probability density of obtaining a given $\mathbf{R}$ conditioned on $\mathbf{r}$ (see Supplementary Information). Similar to $\mathbf{M}$, $\kappa$ is a modeling choice selected before training that defines the multiscale relationship between resolutions. Although arbitrary $\kappa$s can be used, in this work we focus on $\kappa$s defined by probabilistically extending a given CG mapping $\mathbf{M}$. Specifically, we let $\kappa$ follow an isotropic Gaussian distribution centered around $\mathbf{Mr}$:

$$\kappa(\mathbf{R}, \mathbf{r}) \propto \exp[-(\mathbf{Mr} - \mathbf{R})^\mathsf{T} \Sigma^{-1} (\mathbf{Mr} - \mathbf{R})/2], \qquad (5)$$

where the covariance matrix $\Sigma$ is defined to be $\sigma^2 I$ with variance $\sigma^2$ referred to as the noise level; $\sigma$ is a hyperparameter that improves training on finite datasets but introduces bias into the training objective (see Supplementary Information). Practically, for given $\mathbf{r}$, $\mathbf{R}$ can be sampled by first applying the map $\mathbf{M}$ and then adding scaled standard Gaussian noise. Training consists of minimizing Eq. (3) by repeatedly drawing atomistic MD simulation frames, generating random noised counterparts by conditionally sampling $\kappa$, combining saved force and generated noise information via Eq. (4), and subsequently performing gradient-based optimization to minimize the prediction error in Eq. (3) (see Supplementary Information). Although this approach has

fundamental connections to DDM training, no DDM is directly used in this procedure.

In Eq. (1), the force map $\mathbf{T}$ projects the atomistic forces into the CG space. Different choices of $\mathbf{T}$s can be made for a given choice of CG coordinates[36] (see Supplementary Information). In the context of Eq. (3), $\mathbf{T}$ also specifies how to combine distributional and atomistic force information. $\mathbf{T}$ may be defined to include only contributions from $\kappa$, facilitating force-field optimization when no atomistic forces are available; the corresponding $\mathbf{T}$ is referred to as $\mathbf{T}_{noise}$. In the case of $\mathbf{T}_{noise}$, training in the proposed framework is equivalent to the score-matching training of a DDM: Eq. (3) reduces to Eq. (2) at a fixed noise level. On the other hand, Eq. (3) reduces to Eq. (1) if a $\kappa$ concentrated at $\mathbf{Mr}$ is considered.

Critically, however, $\mathbf{T}$ may instead be defined to include contributions from both saved atomistic forces and noising, allowing the seamless combination of distributional and force-based information during training. Preliminary work has shown that combining reference force information with noising may improve performance in the low-noise stage of DDMs in two dimensions[54]; the current work demonstrates that the theory of bottom-up CG can unify this information to create state-of-the-art MLCG force fields on complex systems with drastically reduced amounts of training data while retaining a rigorous connection to the atomistic system.

**Improved MLCG force-fields.** We demonstrate the performance of MLCG force fields trained to minimize Eqs. (1 or 3) for Chignolin[46], Trp-Cage[47], NTL9 (K12M mutant)[48], and Ubiquitin; results for Chignolin are provided in the Supporting Information.

Chignolin, Trp-Cage, and NTL9 have previously required millions of frames for accurate MLCG force-field training[22,55]; Ubiquitin has additionally required that these training data be drawn from simulations of multiple biological systems[23]. Force-fields were created at a CG resolution of one site per $\alpha$-carbon and trained using batched gradient-based optimization. When minimizing Eq. (1), batches were used without modification to estimate the gradient. However, in the case of Eq. (3) with a kernel defined in Eq. (5), batches were drawn and augmented with standard Gaussian noise. This procedure is described in Fig. 1. When minimizing Eq. (1), $\mathbf{T}$ is then defined to minimize $\langle \|\mathbf{T}f\|^2\rangle$ as in previous work[36] (see Supplementary Information). When minimizing Eq. (3), two $\mathbf{T}$ were used: the first combined information from both atomistic and noise-derived forces from the minimization of $\langle \mathbb{E}_\kappa \| \mathbf{T}f\|^2\rangle$ (see Supplementary Information for details, in particular Fig. S1), and the second used only noise-related information, $\mathbf{T} = \mathbf{T}_{noise}$.

**Trp-Cage.** MLCG models of Trp-Cage[47], a 20-residue miniprotein, were optimized using training sets of varying sizes and $\kappa$s of different variances (different noise levels) to investigate the data efficiency of the proposed learning procedures. Training samples were extracted from adaptively sampled short atomistic MD simulations[22] at $350K$ that do not distribute according to the many-body potential of mean force (PMF); for model validation, this data was reweighted using a Markov State Model (MSM)[56]. Model accuracy was quantified using low-dimensional free energy surfaces (FESs) generated by simulating the CG force-field using unbiased MD. FESs were calculated along low-dimensional projections capturing the folding-unfolding process as revealed by TICA[57] on the atomistic trajectories (Fig. 2). In Fig. 2a, b, we focus on the distribution on the first TIC dimension resolving the folding-unfolding transition, while the quantitative comparisons in Fig. 2c are made over histograms on the first two TICs. An analysis of the fraction of native contacts shows that the formation order of different structural motifs are well recovered by the models, demonstrating an application in rare-event sampling of folding-unfolding transition states (see Supplementary information IIE). We also include in the Supplementary Information the two-dimensional FESs for

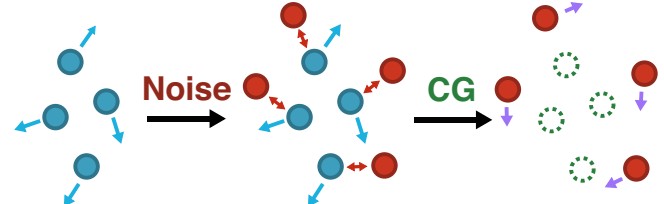

**Fig. 1 | An illustration of the proposed training strategy.** Training data consisting of "real" particles with associated forces (blue) are first combined with noise to add additional sites (red); these new particles interact with the original particles, changing forces throughout the system. The "real" particles (green dotted circles) are then systematically CG out and associated with a linear combination of "real" and noise-derived force information (purple), providing data for subsequent force-field training.

models and references over TIC1–TIC2, as well as RMSD–$R_g$ (radius of gyration).

Models trained on atomistic forces without noise using Eq. (1) recover the reweighted FES (despite the biased training set) with sufficient training data (1.6M samples, Fig. 2a, left panel). In contrast, minimizing Eq. (3) using only noise information produces samples biased towards the distribution of training data (Fig. 2a, left panel). In this high-data regime, combining these sources of information by minimizing Eq. (3) with an optimized $\mathbf{T}$ results in similar accuracy as traditional force-matching; however, this approach greatly increases performance at lower training set sizes, yielding significantly more accuracy than traditional force-matching when fewer than 1M samples are available. Strikingly, training with only noise information is as accurate (for noise level, i.e. variance, $5 \times 10^{-4}\ \text{Å}^2$), or even more accurate (for noise level 0.003 and 0.005 Å$^2$), than conventional force-matching (Eq. 1) in this low data regime (Fig. 2a, middle; c, right panel); moreover, this data efficiency is further increased by including force information. We note that high noise levels ($>0.01\ \text{Å}^2$) can be detrimental, perhaps due to breakdown of the prior energy terms in MLCG training (see Supplementary Information for details).

**NTL9.** The proposed training methodologies were additionally validated on NTL9 (K12M)[48], a 39-residue protein. As in the case of Trp-Cage, models were optimized using training sets of varying sizes and $\kappa$s of different variances (different noise levels). NTL9 represents a significantly more challenging learning target than Trp-Cage, featuring a slower folding-unfolding transition and several folding intermediates[15,16,58] (Fig. 3d). Training samples were similarly extracted from adaptively sampled short atomistic MD simulations[22] at $350K$ that do not reflect the many-body PMF; in this case, creation of a converged MSM was not feasible (see "Methods" section). Instead, millisecond-length trajectories generated at $355K$ by D.E. Shaw Research (DESRES) using the Anton computer with the same force-field and molecular topology were utilized as an equilibrium reference[16]. The distribution of samples differs significantly between the training data and the reference Anton simulations (Fig. 3a, e). We note that, as revealed by refs. 58,59, even these long Anton simulations may deviate from the underlying many-body PMF for minor states.

Similar to Trp-Cage, models trained using traditional force matching and a combination of force and noise information accurately approximate the distribution of the reference data despite the biased training data; in comparison, those trained using only noise information are biased towards the training distribution (Fig. 3a). We note that models trained using only noise information still exhibit a bias towards the correct equilibrium distribution, while in principle should reproduce the distribution of the training data. This bias may be induced by the CGSchnet[20] architecture (see the "Discussion" section). At lower training sizes, the combination of force and noise information greatly increases accuracy. Interestingly, at intermediate noise levels, the

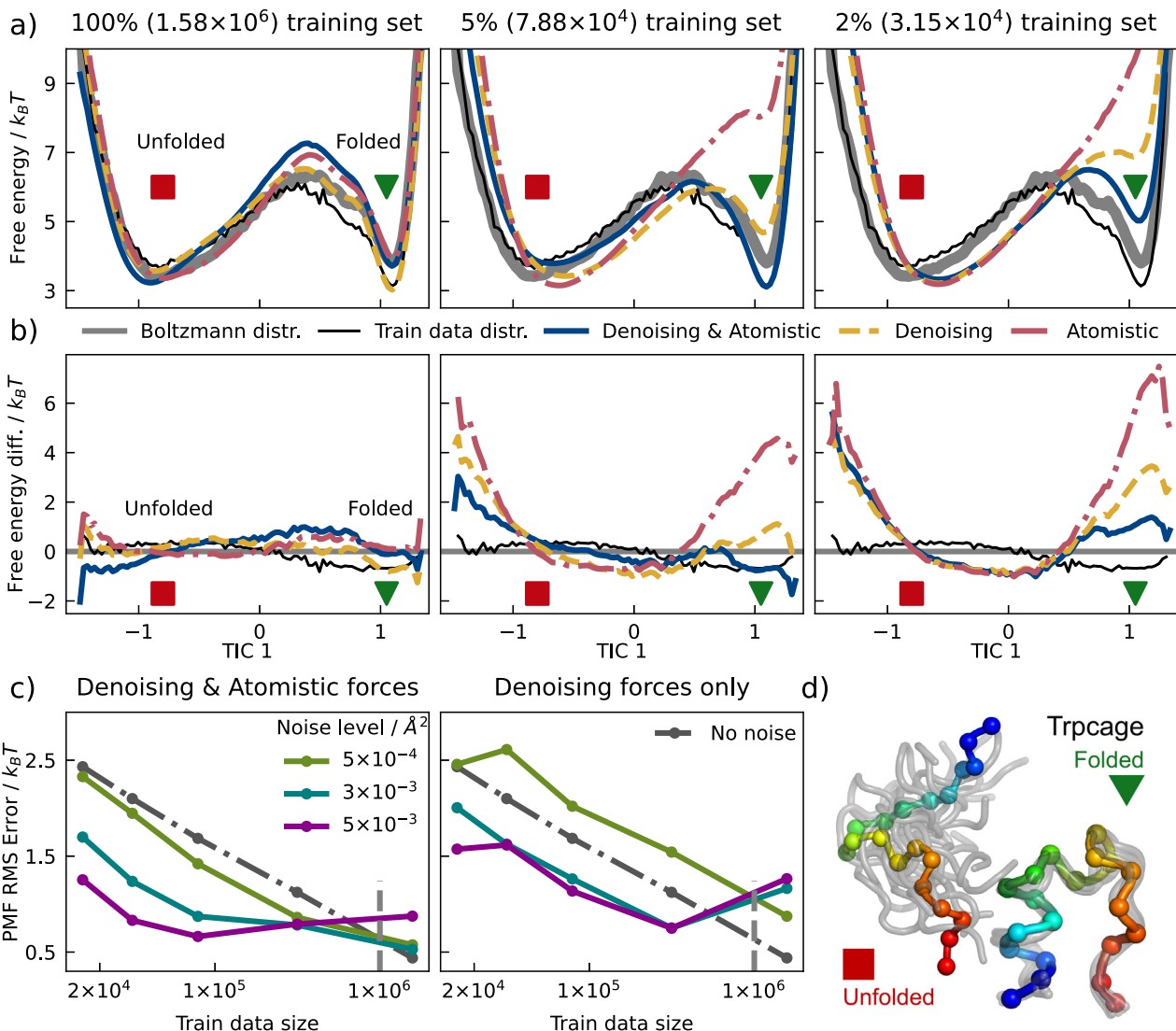

**Fig. 2 | MLCG models of Trp-Cage. a** The 1-D free energy over the first time-lagged independent component (TIC) for models trained with a combination of denoising forces and mapped atomistic forces (dark blue), with denoising forces (yellow), or with atomistic forces (red) on different strides of the training dataset. The Markov-State-Model (MSM) reweighted reference surface is shown in solid gray and serves as a proxy to the Boltzmann distribution, and training data distribution in thin black. Only a single noise level (0.003 Å²) is visualized. **b** Difference of 1-D free energy over the first TIC for models and training dataset when compared with the MSM-reweighted reference. The color scheme is the same as subplot (**a**). **c** Dependence of model accuracy on training size at different noise levels. **d** Coarse-grained representation of Trp-Cage, where the red square and green triangle represent the unfolded and folded metastable states, respectively. Source data are provided as a Source Data file.

combined model can produce a reliable MLCG force-field with only 1% of the whole training set (76k frames, bottom left of Fig. 3c). We note that the PMF error saturates at $7.5k_BT$ for the other models (noise only or forces only) at lowest training set sizes, as this corresponds to a model completely destabilizing the folded state and folding intermediates (Fig. 3e). At an intermediate noise level, the model is also capable of reproducing the majority of the native contact trends (see Supplementary Information).

**Ubiquitin.** To demonstrate the applicability of the proposed training procedure to larger biomolecular systems, Ubiquitin (PDB: 1D3Z, 76 residues) was also modeled. Simulating the equilibrium folding/unfolding process of Ubiquitin at 300K has remained a longstanding challenge in the simulation community[60,61] due to the high stability of the folded state. For example, even on the Anton specialized super-computer, an elevated temperature (395 K) was required[61] to observe

folding and unfolding transitions, with total simulation lengths reaching more than 7 ms.

Here, we employ the proposed approach to simulate equilibrium folding/unfolding of Ubiquitin with a $C_\alpha$ MLCG trained on short out-of-equilibrium atomistic simulations. Training data were obtained from atomistic MD performed at 300K using the AMBER 99SB-ILDN force field[62]; these simulations were performed using two setups. First, short unbiased simulations initialized from the crystal structure were used to characterize the fluctuations present in the folded basin. Second, constant-velocity steered MD was used to unfold the protein by applying a stretching force to both termini; this procedure forced the simulation to sample conformations where the hydrophobic core of the protein was exposed. These steered MD simulations were performed using an aggressive velocity rate (0.69 nm/ns) and cannot be reweighted to obtain the equilibrium distribution of the system; as a result, we do not have a reference free energy profile from atomistic

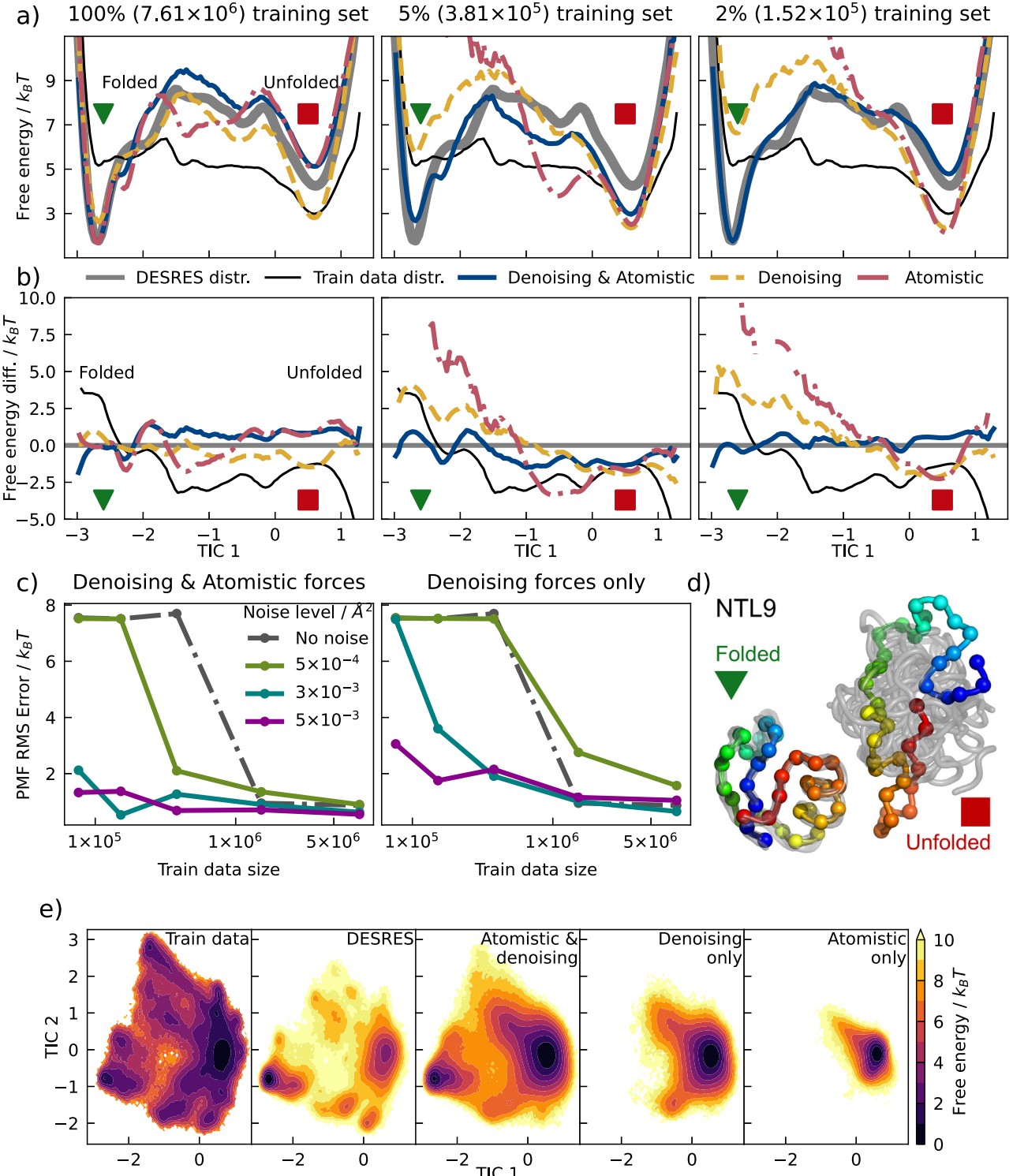

**Fig. 3 | MLCG model of NTL9. a** The 1-D FES over the first TIC for models trained with a combination of denoising forces and atomistic forces (dark blue), with denoising forces (yellow) or with atomistic forces (red) on different strides of the training dataset. The reference FES estimated from DESRES simulations[16] is shown in solid gray, and the training data distribution in thin black. Only noise level 0.003 Å² is shown. **b** Difference of 1-D free energy over the first TIC for models and training dataset when compared with the reference. The color scheme is the same

as subplot (**a**). **c** Dependence of model accuracy on training size and noise levels. **d** Coarse grained representation of NTL9 in its folded (green triangle) and unfolded states (red square) corresponding to the symbols in subplots (**a**, **b**). **e** 2-D FESs for NTL9 reference simulations and MLCG models trained with 1% of the training data. Noise level is 0.003 Å² when noise information is used. Source data are provided as a Source Data file.

simulations to which to compare our CG model. The accelerated pulling rate allowed the unfolding of the protein with steered MD simulations in 230 ns, which could be accomplished in less than 3 weeks using a single Nvidia-RTX3090 GPU. The total size of the

training dataset, consisting of short folded simulations and steered MD simulations, contained 250,000 instances of configurations and forces, distributed as illustrated in Fig. 4a (leftmost panel). Additional details are provided in the "Methods" section.

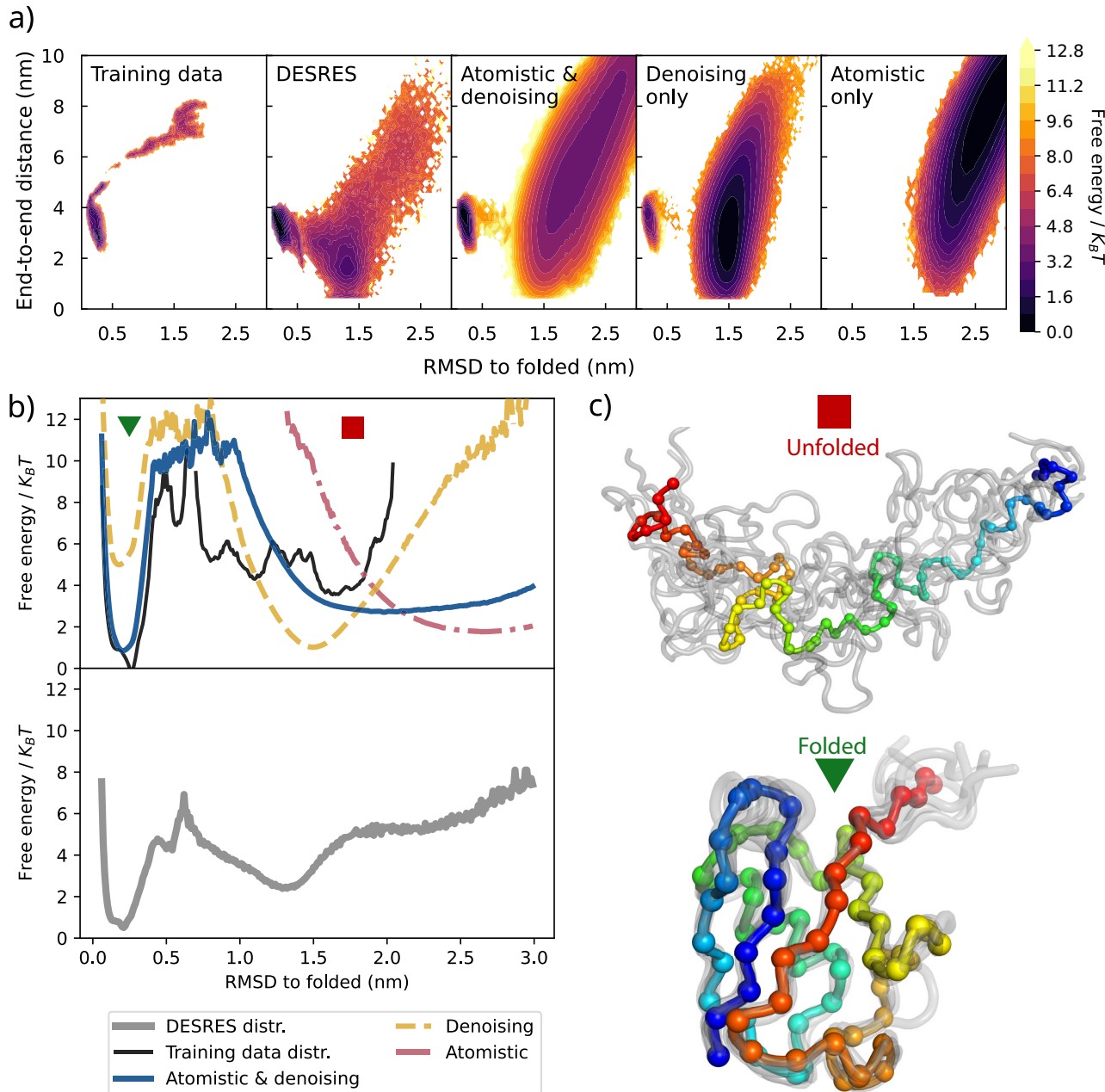

**Fig. 4 | MLCG model of ubiquitin. a** 2-D FES obtained from parallel tempering simulations for different models, the (biased) training data, and the reference D.E. Shaw Research (DESRES) simulations[61]. The force-only model is unable to stabilize the folded state. **b** 1-D FES on Cα-RMSD (root mean square deviation) to the native state, which shows that the denoising-only model biases towards unfolded states over native fold. The FES corresponding to the DESRES simulations is shown separately as it has a different temperature (395 K) and a difference force-field. **c** Ensemble of structures from folded and unfolded basins of the atomistic & denoising model. Source data are provided as a Source Data file.

Several MLCG models for Ubiquitin were trained on this out-of-equilibrium atomistic dataset, and for each model, equilibrium FESs were estimated using parallel tempering (see Supplementary Information IIB). Figure 4 shows that the folded state is not sampled using the force-only MLCG model—it is fully unfolded. In contrast, MLCG models trained on either atomistic and denoising or denoising-only forces both possess metastable folded and unfolded states. However, only the model trained with a combination of atomistic and denoising forces yields the folded state as the most stable state, as shown in Fig. 4b. It is interesting to note that although both atomistic and denoising and denoising-only models exhibit high folding barriers, neither model visits the highly distorted extended

states that were generated by steered MD and are present in the training data.

The only existing folding/unfolding atomistic simulation of Ubiquitin is the aforementioned one performed by DESRES on the Anton supercomputer at 395 K, and with a force-field (CHARMM22*[63]) different from ours. The behavior of our trained MLCG models can thus be compared only with the Anton simulation on a qualitative basis. We also note that the DESRES simulations visualized in Fig. 4b (lower panel) were performed with a periodic box length of 5.8 nm; as a result, it is unclear whether the unfolded basin is affected by self-interaction artifacts. In contrast, the steered MD simulations used to generate the training data employed a box length of 15 nm (see the "Methods" section).

## Discussion

MLCG MD represents a promising alternative to atomistic simulation[8,23]. However, training represents a significant barrier to applying MLCG force-fields to biomolecules[21]: Existing approaches have required large amounts of training data[23,36], repeated converged simulations of the MLCG model being trained[50,51], or carefully modified sampling of the atomistic system under study[33,38]. These difficulties have significantly hampered the development of MLCG models relative to their atomistic counterparts[21,64].

The present work represents a significant step forward by substantially reducing the amount of data required to efficiently train biomolecular MLCG models on atomistic MD trajectories. When force information has been recorded during atomistic MD, the proposed method significantly improves on the data efficiency of traditional force matching while retaining its ability to correctly infer the underlying Boltzmann distribution from unconverged simulations. In the absence of force information, the proposed approach targets the sampled distribution with higher data efficiency than traditional force matching. In both cases, no simulation of the MLCG force-field is required during optimization, representing a significant improvement over previous distribution-based training approaches[50,51]. As demonstrated by the Ubiquitin example, these advances enable MLCG models of biologically relevant proteins to be created using consumer-grade resources and without the need for exceedingly long equilibrium atomistic simulations. The proposed methodological advances have been developed in a publicly available code base.

Despite the demonstrated improvements in accuracy, it is not yet theoretically clear how the addition of configurational noise ($\kappa$) stabilizes training on finite datasets. Atomistic forces from individual configurations provide a noisy source of reference information, and previous work has shown that reducing this force noise improves training[36]. In this regard, the introduction of configurational noise may have two possible impacts. First, the distributional information contained in the "noise forces" represents independent training information and may therefore give rise to a less noisy reference signal for optimization when combined with atomistic forces. Second, Eq. (3) can be viewed as combining information across nearby configurations (Eq. S2), possibly reducing the impact of atomistic force fluctuations. However, additional research is needed to elucidate and extend the improvements shown in this article. Additional reasoning and discussions about the role and choice of $\kappa$ are provided in the Supplementary Information.

Multiple MLCG force fields presented in previous sections were trained on non-equilibrium data, but produced FESs that more closely resembled equilibrium distributions rather than their training dataset. When training on atomistic forces, this may be due to the information contained in the forces themselves. However, when training solely on noise-derived force information, the training residual by definition drives the model towards the empirical data distribution, not the equilibrium distribution. We hypothesize that the inherent inductive bias in the neural network (CGSchnet[20]) plays a dominant role in the resulting FES when sufficient training data are present; here, this bias results in equilibrium-like distributions. While in principle architectural bias impacts both noised and non-noised training, it seems likely that non-noised training requires substantially more training data to approach this error regime, and we believe that the reduced data requirements demonstrated for the proposed techniques provide a foundation for future investigations into more expressive architectures.

The proposed training approach is based on a well-defined reinterpretation of the theory underpinning systematic CG[28] and DDMs. This mathematical framing modifies traditional CG to incorporate a probabilistic connection between the fine and coarse resolutions (Eq. S2), allowing for the transparent application of existing CG theory, such as thermodynamic analysis, to the resulting models[8,9]. These connections may additionally provide avenues for improving the efficiency of DDMs focused on conformational distributions[39]. Since the relationship between the derived forces and distribution of configurations remains valid, the proposed improvements may be directly combined with other approaches for MLCG force-field training[22,32–35,38,65].

The addition of configurational noise may impact other tasks central to CG MD. For example, substantial work has recently focused on reintroducing atomistic details into CG protein configurations using ML-based techniques[66–74]. Many approaches produce atomistic configurations that unambiguously map to their CG counterparts, consistent with traditional bottom-up CG; however, this same reconstruction task naturally incorporates additional flexibility between a given atomistic and CG configuration when interpreted with the proposed probabilistic map. Although this formulation has previously appeared as a practical relaxation for numerical optimization (e.g., refs. [69,70]), it can instead be viewed as an exact solution in this probabilistic setting. This compatibility between reconstruction and the proposed resolution coupling may, in turn, improve methods that directly utilize said reconstruction to accelerate conformational sampling. In this context, recently, Zhang et al.[69] proposed learning the fine-coarse resolution coupling from data, and using the learned transformation in a generative fashion for efficient sampling in an active learning scheme. However, we note that the full application of these methods to explicit solvent molecular systems often requires complete reconstruction of the all-atom resolution, including the placement of water molecules for each protein conformation according to the Boltzmann distribution. To our knowledge, such a reconstruction is still absent from typical procedures. In contrast, the focus of the present work is on the efficient training of machine-learned coarse-grained models in a data-sparse regime without reference to reconstruction; the learned effective energy function for the CG system can then be used with physics-based methods (e.g., MD) to sample the conformational landscape and estimate free energy differences.

Many questions about MLCG force fields remain open. While unconverged, the training data used in this study captured a substantial level of conformational diversity; as a result, it remains an open question how the proposed methods may perform on data with limited diversity, e.g., whether a model can recover states absent from the data. Furthermore, while the proteins considered in this work are challenging targets for MLCG force-fields[22,39], they do not represent a biologically relevant application; the effect of the proposed procedures on complex CG force-fields[23] remains to be investigated. Nevertheless, the proposed approach represents a striking gain in data efficiency over previous work[22], greatly increasing the applicability of a promising class of physically informed ML models.

## Methods

### Optimizing CG force-fields

MLCG force fields were optimized to minimize their force prediction error on CG conformations. The CG resolution was defined by the removal of all atoms except the alpha carbons of each amino acid; this is algebraically encoded in $\mathbf{M}$ or $\kappa$, depending on the training equation. CGSchNet[20] (a modified SchNet[75] graph neural network plus prior energy terms) with two interaction blocks was used to represent all CG force-fields[20]. Auto-differentiation was used to obtain the forces on each CG bead.

When optimizing force-fields using traditional force matching, established procedures were used[19,20,36]. Positions and forces were drawn from atomistic MD simulations and mapped to the CG resolution using preselected $\mathbf{M}$ and $\mathbf{T}$ matrices and used to generate batched gradient updates using the ADAM optimizer.

When minimizing Eq. (3), the gradient update was modified as follows. Batches were first drawn and mapped to the non-noised alpha carbon resolution. Second, 0-mean Gaussian noise with diagonal

covariance was added to the positions; collectively, this corresponds to $\kappa(\mathbf{R}, \mathbf{r})$ defined to be a Gaussian centered at **Mr**. Third, forces were modified according to Eq. (4) with a preselected **T**. **T** was defined either to minimize $\langle \mathbb{E}_\kappa \parallel \mathbf{T}f \parallel^2 \rangle$ or, in the case of noise-only forces, directly defined via Eq. (S4) (see Supplementary Information).

To investigate the data efficiency of different training strategies and noise levels, the training and validation set size was varied. Smaller data sets were created from the full data set via striding, ensuring that the conformational distribution did not change. For each training condition, a representative model was selected for MD analysis using the following criteria. After the initial decrease in the validation loss, if a consistent increase was observed over multiple epochs, the model at the lowest reached validation loss was selected. In case of no obvious increase (e.g., for large training sizes with denoising information), training was stopped at a certain number of epochs (e.g., 200 for NTL9 models on full training data) and used for simulation analyses. Note that there were cases where the model quality seemed to be sensitive to stochastic fluctuations during training or deteriorated over epochs despite validation losses continuing to gradually decrease, especially for NTL9 models at intermediate noise levels (e.g., for a level of 0.003 Å$^2$). If simulation instability was observed, an earlier epoch was selected. More details can be found in the Supplementary Information.

## Optimizing T
All force maps were optimized using the aggforce package[76] (version `1.0.1`); force maps including noise were generated using the `stagedjoptgauss_map` and `stagedjslicegauss_map` methods. Supporting libraries include numpy[77] (version `1.26.4`), jax[78] (version `0.4.35`), pandas[79] (version `1.15.2`), scipy[80] (version `1.1`), and OSQP[81,82] (version `0.6.7.post3`). Force maps were optimized using an L2 regularization of $10^3$ for atomistic contributions and 5 for post-map noise contributions (see Supplementary Information); forces were considered in units of kcal/(mol · Å). Force map optimization was performed using 37,302, 350,000, 500,000, and 784,000 MD frames for Chignolin, Trp-Cage, NTL9, and Ubiquitin, respectively.

## Reference atomistic MD simulations
MLCG models were trained and validated against solvated all-atom MD simulations. For training Chignolin, TRPcage, and NTL9 models, a dataset generated on GPUGRID[83] was used[22], which utilized thousands of short simulations seeded with an adaptive sampling strategy. Collectively, this dataset provides approximately 1.8 M frames of coordinates and forces from 3744 simulations (~50 ns each) for Chignolin, 2 M frames from 3940 simulations (50 ns each) for Trp-Cage, and 9.5 M frames from 47,599 trajectories (20 ns each) for NTL9. The simulation procedure and data availability are described in ref. 22. Owing to the long timescale of major conformational transitions, the data distribution deviates from the true Boltzmann distribution for the all-atom systems. For Trp-Cage, we obtain the reference FES with MSM reweighting as in ref. 36. For NTL9, due to difficulties in obtaining implied timescales consistent with Markovian dynamics[56], data generated using the Anton supercomputer by D. E. Shaw Research was used as a reference for the equilibrium distribution. This data comprises four long simulations (2.9 ms in aggregate time) with the same force-field and solvation setup[16]. The Anton simulations were performed at a temperature of 355K instead of 350K of the training data from GPUGRID, implying that the folded state is expected to be slightly destabilized relative to the targeted many-body potential of mean force.

For ubiquitin, a dataset was generated using the AMBER ff99SB-ILDN force field[62] at 300 K. To sample fluctuations around the folded state, 28 independent trajectories were generated, all of them starting from the reference crystal structure of the native state, for a total of 158 μs of simulation time; these trajectories were qualitatively identical. To obtain conformations further away from the folded basin, a single trajectory was generated using steered MD by applying a force constant of ~120 kcal/(mol nm$^2$) to the alpha carbons of the first and last residues of the protein. This pulling force was used to stretch the protein from the end-to-end distance of 3 nm to 8.15 nm. The pulling force was applied as a harmonic restraint over the end-to-end distance for 115 uniformly spaced partitions of the mentioned interval. For each of these windows, 4 ns of simulation were run with data recorded for only the last 2 ns. In total, the steered MD dataset amounts to 230 ns of simulation time. We note that the DESRES simulations of Ubiquitin[63] cannot be used for training as they do not contain forces.

For the atomistic Ubiquitin simulations, a non-bonded cutoff of 9 Å and switch distance of 7.5 Å were applied. Long-range electrostatics were handled using the Particle Mesh Ewald method, and H-bonds were constrained. The system was constructed using a cubic box and solvated using TIP3P water and Na$^+$, Cl$^-$ ions for a salt concentration of 0.1 μM. For unbiased folded simulations, the side of the box was set to 6.5 nm, creating a system with 26,784 atoms. For the steered MD simulations, the size was increased to 15 nm to account for the full unfolding of the protein, which produced a system with 325,004 atoms. After preparing the solvated boxes, the energy of the system was minimized, and then 8 ns of NPT equilibration was performed using Monte Carlo Barostat and Langevin dynamics with a friction coefficient of 0.1 ps$^{-1}$ and a 2 fs timestep. After equilibration, production simulations were performed using a Langevin dynamics with a friction coefficient of 0.5 ps$^{-1}$. The preparation of the systems was performed with GROMACS[84] (version `2022.5`) while the equilibration and production simulations were performed in OpenMM[85] (version `8.0.0`).

## CG MD simulations
Trained MLCG models of Chignolin, TRPCage, and NTL9 were evaluated by performing 100 MD simulations at the same temperature as the training data (350 K). The starting structures for simulation were randomly sampled from the training set. The simulations used the Langevin thermostat with a friction coefficient of 1 ps$^{-1}$ and a timestep of 2 fs. For Chignolin, Trp-Cage, and NTL9, each trajectory was simulated for 2 M, 5 M, and at least 10 M time steps, respectively. The simulation time was sufficiently long to allow for several transitions along the first TIC. Note that a small number of NTL9 models exhibited integration instability; this was addressed by re-initialization of the corresponding simulation. This might be attributed to the imperfect functional form and fitting of prior terms based on Gaussian-noised statistics. For details of the implementation, we refer to ref. 36 and Supplementary Information.

Ubiquitin models were evaluated using both Langevin dynamics and PT simulations; the Langevin simulations were unable to converge the corresponding FES within 20M timesteps due to a high transition barrier. Using a set of 10 geometrically-spaced temperatures between 284.1 and 362 K and an exchange interval of 4 ps, PT simulations were initialized from 20 different structures, including configurations from the folded basin and the transition region (Fig. S5). The observed exchange acceptance rate for all temperatures was higher than 0.3[86]. PT Simulations ran for at least 13 M time steps using a Langevin thermostat with a friction coefficient of 1 ps$^{-1}$ and a timestep of 4 fs.

## Data availability
The all-atom MD simulations used for the model training for Chignolin, TRPcage, and NTL9 are the same as in ref. 22 and are publicly available at https://zenodo.org/records/8155343[87]. In-house simulations for Ubiquitin and all the trained models presented in this study are publicly available at https://zenodo.org/records/18756638[88]. Reference DESRES simulations from existing publications[16,61] were requested from D. E. Shaw Research and served only for comparison and are not essential for model training. Source data are provided with this manuscript. Source data are provided with this paper.

## Code availability

Noise-based data transformations were implemented in the aggforce package (version `1.0.1`), which is publicly available at https://zenodo.org/records/18784685[76], and training was performed using the mlcg package (version `noise-alpha`), which is publicly available at https://zenodo.org/records/18770351[89]. A demo showing how to run the code is publicly available at https://zenodo.org/records/18756638[88].

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

## Acknowledgements

The authors thank Andreas Krämer and Atharva Kelkar for helpful discussions and Jacopo Venturin for MSM information. We gratefully acknowledge funding from the International Max Planck Research School for Biology and Computation (IMPRS–BAC), the Bundesministerium für Bildung und Forschung BMBF (project FAIME 01IS24076), the Deutsche Forschungsgemeinschaft DFG (SFB/TRR 186, Project A12; SFB 1114, Projects B03, B08, and A04), the National Science

Foundation (PHY-2019745), and the Einstein Foundation Berlin (Project 0420815101), the computing time provided on the supercomputer Lise at NHR@ZIB as part of the NHR infrastructure, and the computing time provided on the JUWELS supercomputer at the Jülich Supercomputing Centre (JSC).

## Author contributions
A.E.P.D., Y.C., A.S.P.-T., C.C., and F.N. designed research. A.E.P.D. performed the in-house all-atom MD simulations. A.E.P.D., Y.C., and A.S.P.-T. curated the datasets and developed the software. Y.C. and A.S.P.-T. conducted model training, validation, and CG simulations. A.E.P.D., Y.C., A.S.P.-T., C.C., and F.N. analyzed computational experiments. A.E.P.D., Y.C., A.S.P.-T., F.N., and C.C. wrote the manuscript.

## Funding

## Competing interests
The authors declare no competing interests.
