## [Transparent Peer Review file · Nature Communications]

Learning data efficient coarse-grained molecular dynamics from forces and noise

Corresponding Author: Professor Cecilia Clementi

Version 0:

Reviewer comments:

Reviewer #1

(Remarks to the Author)

This article introduced a machine-learning-based coarse-grained force field training approach, which improves the MLCG training by combining force-based and noise-based learning. This training method is an interesting approach, but in my review of related works of literature, I found a similar work in arXiv preprints first submitted in May 2023: arXiv:2305.01243v1, which proposes a method called cycle coarse-graining (CCG). This preprint article claimed that CCG can perform both coarse-graining (CG) and fine-grained reconstruction (FCR) operations. Both approaches use generative learning, but their specific implementations are different. The noise-based generative learning proposed in this paper is more accessible to implement than the generative adversarial network (GAN) on which the CCG is based. However, the CCG method can perform FCR in addition to CG. As a different CG approach, the method proposed in this article still has value for publication. However, since a similar approach has already been posted, this article should be majorly revised to compare it with the CCG method systematically.

1. The authors should systematically compare their approach to generative learning with the CCG. In particular, the loss function of the method should be compared with that of the CCG (Equation S7 in arXiv:2305.01243v1) to see if there is a fundamental conceptual difference between the two. In addition, the authors could also discuss the advantages and disadvantages of the two approaches in generating learning.
2. Is it possible for the method to do FCR in some way like the CCG?
3. The method is claimed to be a "data efficient" method, and the authors of the CCG also claim that their method can be trained with fewer samples. Can the authors compare the difference in training samples between the two methods? Who can use less data to train a CG force field with the same system and the same accuracy?
4. The arXiv:2305.01243v1 article trained the CG force field of the peptide Chignolin. Could the authors also train the force field of the same system for comparison? In addition, when demonstrating the effect of a peptide CG force field, it would make more sense to project the free energy on some physically meaningful reaction coordinates, such as Rg and RMSD, as the CCG article does, rather than on the TICA coordinates. FES of the already trained Trp-cage and NTL9 force fields should also be drawn this way.
5. The CCG approach also claimed to solve the rare event sampling problem through active learning. They trained a CG force field for chemical reactions in addition to the protein force field. Can the authors indicate whether the methodology of this paper can be easily trained for such systems involving rare events?

In summary, this paper is methodologically innovative as a new generative learning-based approach to coarse-grained force field training. This article is worthy of publication, but before doing so, it should be systematically compared with already released methods.

(Remarks on code availability)

Reviewer #2

(Remarks to the Author)

This manuscript studies the effect of adding a noise bias to the widely force-matching objective function for bottom-up coarse-grained modeling. The authors explore numerically on two small peptides how the quality of the many-body potential of mean force (mb-PMF) is represented with force mapping procedures that are deterministic or noisy with a particular focus on the size of the data set used to train the coarse-grained model. They draw analogies to denoising diffusion models but I do not think there is yet a strong theoretical justification for this empirically powerful approach.

Overall, I think this is a nice contribution to the large literature on building coarse-grained models and would be happy to see it published in its current form. The results are quite compelling: the numerical results demonstrate unambiguously that the representations of the mb-PMF improve with appropriate noising of the training data and furthermore, the noising procedure massively reduces the total amount of data required to build a CG forcefield. Despite the fact that the configuration space is well-sampled on these data rich problems, the conventional force matching objective does not perform anywhere near as well as the noised objectives.

I have a couple suggestions that I believe could improve the presentation of the paper, which I would like the authors to consider:

- How the loss in (3) is evaluated is not well-described in the text. It would be beneficial to explain the ancestral sampling procedure shortly after equations (3) and (4) because it's an important aspect of actually using this objective. In particular, the procedure for evaluating (4) of $f(r,R)$ is not clear in the main text, though well explained in the appendix.
- The fact that the distribution κ a modeling choice with hyperparameters also merits a clear discussion. I feel that previous work from the authors on optimal aggregation of forces could be better leveraged to explain the additional noise as accounting for uncertainty in the projected forces.
- In Fig. 2, the caption mentions that a single "noise level" is visualized and that term appears throughout the text interchangeably with "noise variance". While I think it's clear from the units what the quantity means, it should be defined and explained somewhere.
- For the coarse-grained simulations, it is mentioned that 100 trajectories are collected, but where the initial conditions are obtained is not explained. Do these also have good coverage of the mb-PMF?

(Remarks on code availability)

Reviewer #3

(Remarks to the Author)

In this paper from Noe and Clementi labs propose a hybrid approach that combines denoising score matching (used in diffusion models) and multiscale coarse graining (force matching) to improve training efficiency in CG force fields while reducing the need for large datasets. This method has been tested on proteins such as Trp-Cage and NTL9, demonstrating a significant reduction in data requirements by a factor of 100 while maintaining accurate simulation results. Their model can generate accurate free-energy surfaces and simulate the folding-unfolding process of proteins even with minimal training data. The authors tested the performance with different noise levels and varying amounts of training data, showing that including noise information improves accuracy, especially when data is scarce. The paper is well written and applying the diffusion plus noise augmentation is the innovative parts. I think also the paper might be more suitable for specialized journals such as JCIIM or JCTC. Here are the comments and concerns I have:

1. Obviously, the biggest concern is the generalizability and the ad-hocness of the method to the protein of interest. While the method works well for the tested proteins, its scalability to even more complex biomolecules (or different types of molecular systems) remain to be fully explored. I strongly suggest pre-training on multiple small proteins and test on a completely hold out protein to see the real effect of denoise pretraining. I mean for example pre-train on TRP-cage and NTL9 and test on the Ubiquitin or a small protein. You can fine-tune with a few first step of trajectory (and that can be an ablation study).
2. Authors should explain and clarify the inference time for DDM. Diffusion models are notoriously slow on inference specially for high quality generation and when number of denoising steps go up. A study to show this time is not adding more compute time as opposed to the reduction of the compute time authors pitched for the paper.
3. The reliance on pre-trained atomistic simulations, although reduced, is still a limitation in terms of accessibility for users without high-performance computational resources. For general proteins and specially larger ones such as receptors, such training data (as you obtained from DESRES) might not be available. Again, as I mentioned in the first comment, I would love to see the transferability to larger proteins and different proteins.
4. I have a suggestion, which might be a bit out of scope but makes the work stronger: How about other pretraining techniques such as mask pretraining or bond/atom deletion augmentation? It would be interesting to explore those pretraining strategies.
5. In equation 4 and for total loss, I would like to see a regularization factor (Λ) to see the effect of MD force versus noise.

6. One minor comment: some of the figures in appendix were not compiled properly and are not showing up

(Remarks on code availability)

The code and the results are reproducible.

Version 1:

Reviewer comments:

Reviewer #1

(Remarks to the Author)

In the revised manuscript, the authors' modifications and discussions of relevant details are commendable. The newly incorporated Chignolin system and free energy surface analysis based on Rg/RMSD enhance the comparability of results. Supplementary Figures S7, S9, and S12 in the SI indeed clarify the physical significance.

However, in addressing the core issue I mentioned, that is, the comparison with the CCG method (arXiv:2305.01243v1). There are clear shortcomings, suggesting an attempt to sidestep the crucial points. I explicitly requested a comparison of the loss functions and conceptual differences between the two approaches, yet the authors merely emphasised their avoidance of generative architectures (such as normalizing flows or GANs) without providing a detailed analysis of the fundamental mathematical distinctions in the loss functions. This evasion undermines the article's innovation: merely transplanting denoising score matching without elucidating its unique theoretical contribution to coarse-grained modelling significantly diminishes the methodological value. While the authors cite an article (Nat. Chem. 2025, 7, 1284-1292) as evidence of "extrapolating", they fail to present comparable data in this paper. If the core advantage of noise-assisted training is reducing sample requirements, then a direct comparison should be made between CCG and the present method on identical systems (e.g., Chignolin) to determine the minimum dataset size needed to achieve equivalent accuracy. If the authors cannot obtain the CCG code to evaluate the actual system, they should at least compare the efficiency of the two methods on a numerical toy model.

In addition, regarding the expansion of application scenarios, the authors' response to the rare event sampling problem is equally unconvincing. Yet the authors merely vaguely mention "can be combined with data generated by enhanced sampling" and evade specific case comparisons by citing the lack of all-atom reconstruction methods. In fact, the performance of coarse-grained models in rare events (such as sampling folding transition states) can be evaluated through the statistical properties of their own trajectories, without relying on FCR functionality.

Therefore, the current revision has not yet met my previous modification requirements, particularly failing to clearly delineate the boundaries between this paper's methodology and CCG through quantitative comparisons and theoretical analysis. If these analyses cannot be completed, the assertion of "innovation" should be toned down, explicitly positioning this work as a specific application of denoising score matching within coarse-grained force fields.

(Remarks on code availability)

Reviewer #2

(Remarks to the Author)

I have looked at the response and am happy to see the paper published as is

(Remarks on code availability)

Response to reviewer 1

This article introduced a machine-learning-based coarse-grained force field training approach, which improves the MLCG training by combining force-based and noise-based learning. This training method is an interesting approach, but in my review of related works of literature, I found a similar work in arXiv preprints first submitted in May 2023: arXiv:2305.01243v1, which proposes a method called cycle coarse-graining (CCG). This preprint article claimed that CCG can perform both coarse-graining (CG) and fine-grained reconstruction (FCR) operations. Both approaches use generative learning, but their specific implementations are different. The noise-based generative learning proposed in this paper is more accessible to implement than the generative adversarial network (GAN) on which the CCG is based. However, the CCG method can perform FCR in addition to CG. As a different CG approach, the method proposed in this article still has value for publication. However, since a similar approach has already been posted, this article should be majorly revised to compare it with the CCG method systematically.

We appreciate the reviewer’s kind comments about our work. CCG (arxiv:2305.01243v2) is a novel method for multiscale learning that utilizes normalizing flows trained to link between a fine-grained reference model and a coarser resolution to simultaneously learn how to reconstruct fine-grained samples and approximate the free energy surface at the coarser resolution. This connection allows for multiple new training approaches to be defined. The approach proposed in our manuscript does not directly use any generative machine learning architecture such as normalizing flows, generative adversarial networks, or diffusion models; instead, it extends ideas from denoising score matching (a technique central to diffusion models) to train a neural network force-field at the coarse-resolution. As we do not adopt any generative neural architecture that bridges resolution, we cannot directly compare to many of the results in arxiv:2305.01243v2. We instead here provide discussions on different aspects of the draft such as training residuals and applications.

Reviewer Point P 1.1 — The authors should systematically compare their approach to generative learning with the CCG. In particular, the loss function of the method should be compared with that of the CCG (Equation S7 in arXiv:2305.01243v1) to see if there is a fundamental conceptual difference between the two. In addition, the authors could also discuss the advantages and disadvantages of the two approaches in generating learning.

Author reply: Eq. S7 in arXiv:2305.01243v1 effectively describes a linear combination of 3 loss terms (see S3-S5). The first two terms correspond to losses guiding the reconstruction of atomistic configurations, which we do not perform. The third term corresponds to traditional force matching without the addition of noise. In our manuscript we discuss extensively the connections between our proposed method and traditional force-matching, which is the common term between our loss and the CCG loss. We can not directly compare with the other two terms in the CCG loss, as we do not perform atomistic reconstruction, but we have now noted the utility of our method in the general context of backmapping in the discussion and provided reference to the CCG article (lines 333-347). Other equations presented in arXiv:2305.01243v1 (e.g., S3, a different component of S7) typically focus on methods performing backmapping and often require that configurations at the resolution of the reference energy function be reconstructed. While such terms are difficult for us to use without substantially expanding our method to incorporate backmapping, we also note that the simulations we study as our fine-grained reference systems are at the resolution of fully solvated atomistic configurations

and we are not aware of any backmapping procedure that is capable of effectively constructing a fully solvated atomistic protein structure.

Reviewer Point P 1.2 — Is it possible for the method to do FCR in some way like the CCG?

Author reply: Incorporating fine-grained reconstruction (i.e., backmapping) to our proposed method would be a highly substantial addition to the corresponding manuscript, as the current work does not consider the task of backmapping. While CCG focuses on reconstruction and relates this back to the value of the manybody PMF, our proposed method directly approximates the manybody PMF without reference to any reconstruction. However, we recognize that fine-grained reconstruction is an important topic and we have added a section in the discussion on how backmapping may be affected by the noise addition discussed in the manuscript (lines 333-346).

Reviewer Point P 1.3 — The method is claimed to be a “data efficient” method, and the authors of the CCG also claim that their method can be trained with fewer samples. Can the authors compare the difference in training samples between the two methods? Who can use less data to train a CG force field with the same system and the same accuracy?

Author reply: We have now added Chignolin to our set of test proteins, as in the CCG article. However, we cannot use the same training data as the CCG work as these data do not contain forces on which to train. Conversely, as the code of CCG is not available, we cannot rerun their code on our dataset for comparison.

We further note that, while the current manuscript focuses on reproducing molecular distributions present in the training data, the ultimate utility of the proposed method lies in training models of this type on larger training data distributions in hopes of extrapolating to novel proteins not in the training set. We have demonstrated this in a recent manuscript (Nature Chemistry 17, 1284–1292 (2025)). While the process of estimating the density (e.g., histogramming) of an existing molecular simulation along a low dimensional CV and subsequently combining low-dimensional sampling methods with backmapping can be viewed as a generative model of a given simulation, it is currently unclear how this procedure can be adapted to extrapolate to proteins outside the training set.

Reviewer Point P 1.4 — The arXiv:2305.01243v1 article trained the CG force field of the peptide Chignolin. Could the authors also train the force field of the same system for comparison? In addition, when demonstrating the effect of a peptide CG force field, it would make more sense to project the free energy on some physically meaningful reaction coordinates, such as Rg and RMSD, as the CCG article does, rather than on the TICA coordinates. FES of the already trained Trp-cage and NTL9 force fields should also be drawn this way.

Author reply: Thank you for the suggestion. We have now added results for Chignolin in the SI (Supplementary Information, Lines 301-319) and now provide RG/RMSD FEs in the SI (Supplementary Information, Figures S7, S9 and S12).

Reviewer Point P 1.5 — The CCG approach also claimed to solve the rare event sampling problem through active learning. They trained a CG force field for chemical reactions in addition to the protein force field. Can the authors indicate whether the methodology of this paper can be easily trained for such systems involving rare events?

Author reply: Coarse-grained models trained using force-matching can be used to explore systems that contain rare events. For example, they can be trained on data from smaller, easier to sample reference systems and then used to characterize targets not present in the training set (e.g., our recent Nature Chemistry 7, 1284–1292 (2025) article). Alternatively, force-matched models can be combined with data generated by enhanced sampling and subsequently used to probe rare events. We have now added an example of the latter in the manuscript, by using the learned CG model to study the folding/unfolding transition of Ubiquitin, which is impossible to sample by atomistic simulations at 300 K (even D.E. Shaw Research had to use elevated temperature to simulate it on the Anton computer). Directly utilizing workflows such as that given in arXiv:2305.01243v2, page 11, Fig 4, is only feasible when reconstruction creates samples at the resolution of the reference system. As noted previously, to our knowledge, no such method exists for mapping between coarse-grained and all-atom solvated protein structures, which would correspond to substantial addition to the manuscript.

Response to reviewer 2

This manuscript studies the effect of adding a noise bias to the widely force-matching objective function for bottom-up coarse-grained modeling. The authors explore numerically on two small peptides how the quality of the many-body potential of mean force (mb-PMF) is represented with force mapping procedures that are deterministic or noisy with a particular focus on the size of the data set used to train the coarse-grained model. They draw analogies to denoising diffusion models but I do not think there is yet a strong theoretical justification for this empirically powerful approach.

Overall, I think this is a nice contribution to the large literature on building coarse-grained models and would be happy to see it published in its current form. The results are quite compelling: the numerical results demonstrate unambiguously that the representations of the mb-PMF improve with appropriate noising of the training data and furthermore, the noising procedure massively reduces the total amount of data required to build a CG forcefield. Despite the fact that the configuration space is well-sampled on these data rich problems, the conventional force matching objective does not perform anywhere near as well as the noised objectives.

I have a couple suggestions that I believe could improve the presentation of the paper, which I would like the authors to consider:

Reviewer Point P 2.1 — How the loss in (3) is evaluated is not well-described in the text. It would be beneficial to explain the ancestral sampling procedure shortly after equations (3) and (4) because it’s an important aspect of actually using this objective. In particular, the procedure for evaluating (4) of $f(r, R)$ is not clear in the main text, though well explained in the appendix.

Author reply: Thank you for pointing this out. We moved the most relevant definitions from the Supplementary Information to the main text (lines 121-145), and we hope this point is more clear now.

Reviewer Point P 2.2 — The fact that the distribution κ a modeling choice with hyperparameters also merits a clear discussion. I feel that previous work from the authors on optimal aggregation of forces could be better leveraged to explain the additional noise as accounting for uncertainty in the projected forces.

Author reply: We appreciate the insight of the reviewer. A discussion on this topic has now been included in the Discussion section (lines 307-319), whereas the theoretical reasoning appears in Supplementary Information (lines 85-103). Briefly, the introduction of the noise kernel κ enables

1. Combining information across nearby configurations.
2. Synthesizing the information from the denoising scores and the atomistic forces, which have rather independent sources of uncertainty.

Both effects contribute to the reduction of uncertainty in the final CG forces. However, since the training data distribution (and thus the information from noise forces) is often biased, suitable κ need to be picked like hyperparameter tuning to maximize the benefit from this combination and yet to avoid the discrepancies from the biases.

Reviewer Point P 2.3 — In Fig. 2, the caption mentions that a single "noise level" is visualized and that term appears throughout the text interchangeably with "noise variance". While I think it's clear from the units what the quantity means, it should be defined and explained somewhere.

Author reply: We thank the reviewer for kindly pointing this out. We have included a clear definition of the noise level / extent of noise in Section I(b) (lines 108-110) and mentioned it again in Section I(c) (lines 136-138).

Reviewer Point P 2.4 — For the coarse-grained simulations, it is mentioned that 100 trajectories are collected, but where the initial conditions are obtained is not explained. Do these also have good coverage of the mb-PMF?

Author reply: The starting structures were randomly picked from the training set. We have included this missing description to the Methods section (lines 443-455, 456-458). The sampled structures cover the mb-PMF reasonably well, since most samples are picked from the populated metastable states. Although this might bias the performance of all models towards recovering the mb-PMF, we observed that inaccurate models tend to lead the systems away from the correct states in a small number of time steps, yielding very different probability distributions from the reference.

Response to reviewer 3

In this paper from Noe and Clementi labs propose a hybrid approach that combines denoising score matching (used in diffusion models) and multiscale coarse graining (force matching) to improve training efficiency in CG force fields while reducing the need for large datasets. This method has been tested on proteins such as Trp-Cage and NTL9, demonstrating a significant reduction in data requirements by a factor of 100 while maintaining accurate simulation results. Their model can generate accurate free-energy surfaces and simulate the folding-unfolding process of proteins even with minimal training data. The authors tested the performance with different noise levels and varying amounts of training data, showing that including noise information improves accuracy, especially when data is scarce. The paper is well written and applying the diffusion plus noise augmentation is the innovative parts. I think also the paper might be more suitable for specialized journals such as JCIM or JCTC. Here are the comments and concerns I have:

Reviewer Point P 3.1 — Obviously, the biggest concern is the generalizability and the ad-hocness of the method to the protein of interest. While the method works well for the tested proteins, its scalability to even more complex biomolecules (or different types of molecular systems) remain to be fully explored. I strongly suggest pre-training on multiple small proteins and test on a completely hold out protein to see the real effect of denoise pretraining. I mean for example pre-train on TRP-cage and NTL9 and test on the Ubiquitin or a small protein. You can fine-tune with a few first step of trajectory (and that can be an ablation study).

Author reply: Thank you for the suggestions. We here divide the stated concerns into two separate topics related to the current study: scalability to more complex systems and transferability, e.g., learning from small systems and predicting on larger systems. In the revised manuscript, we have now provided additional experiments for the first topic by directly modeling Ubiquitin using training data drawn from highly unconverged steered MD simulations (lines 237-278). Unlike the previous training datasets, these data were generated using hardware resources available to a typical research group (approximately 3 weeks of simulation using a single RTX3090). While traditional force-matched models cannot stabilize the protein in its native state, the models trained with our proposed method can not only stabilize the correct metastable states, but can also approximate the unfolding order of key structure elements. We believe that this showcases the scalability of the proposed method to larger proteins using resources available to a typical research group.

On the second topic, we have now provided a more detailed discussion in the manuscript on transferability and we summarize it here. Transferable learning is a practical way to create useful coarse-grained models, and the current authors have provided foundational work on how to do so using traditional force matching; this work was recently published in Nature Chemistry (Nat. Chem. 17, 1284–1292 (2025)). As shown in that manuscript, the creation of a transferable model presents multiple substantial engineering challenges beyond the training objective used. Expanding this current article to encompass these challenges would represent a quantity of work beyond what is typical for a Communications article and would dilute its primary content explaining the reasoning behind a novel training objective; we note that the provided numerical examples still represent state-of-the-art performance on the selected molecular systems. We further note that the now-included modeling of Ubiquitin represents an application that is accessible to a typical research group and shows the potential of the proposed approach to go beyond the data used for training. We expect that ultimately the provided results will motivate the incorporation of the proposed training objectives into future transferable modeling attempts.

Reviewer Point P 3.2 — Authors should explain and clarify the inference time for DDM. Diffusion models are notoriously slow on inference specially for high quality generation and when number of denoising steps go up. A study to show this time is not adding more compute time as opposed to the reduction of the compute time authors pitched for the paper.

Author reply: There seems to be a misunderstanding on this point and we apologize with reviewer for not clearly explaining in the original manuscript that no DDM (denoising diffusion model) was used in any experiments in this work. We mentioned DDMs in the manuscript as it is a well-known example of utilizing forces (i.e. scores) derived from a noising process, the same source of information on which our proposed method is built upon. Nevertheless, the methods are vastly different: A DDM aims to reconstruct the data distribution from pure noise-base information by training over many (usually over hundreds of) noise levels, while the proposed models in this work focus on the distribution and the corresponding many-body PMF on a single predetermined noise level. In addition, the sampling processes are distinct: DDMs proceed step-by-step from the highest noise level downward, while our

models are CG potentials and can be used for molecular dynamics (i.e., they do not incur any additional simulation cost relative to a traditional force-matched model). In fact, the computing time required for training is usually shorter for our proposed method than for pure-force-matching MLCG models, as our model reaches high accuracy using a smaller amount of training data. We have now included an explicit discussion on the difference between our approach and a DDM in the manuscript to clarify this matter (lines 144-145).

Reviewer Point P 3.3 — The reliance on pre-trained atomistic simulations, although reduced, is still a limitation in terms of accessibility for users without high-performance computational resources. For general proteins and specially larger ones such as receptors, such training data (as you obtained from DESRES) might not be available. Again, as I mentioned in the first comment, I would love to see the transferability to larger proteins and different proteins.

Author reply: We appreciate the reviewer's consensus with us on the merit of learning a transferable coarse-grained model applicable to different proteins. We have already discussed this topic and its connection with this work in our response to the first point above. Here, we would like to additionally point out that many challenges remain to be overcome before we can reach this "holy grail", and this work addresses the data availability issue by both improving training data efficiency while maintaining compatibility with learning on conformations from unconverged enhanced sampling methods (as demonstrated by our now-provided application to protein Ubiquitin). As the reviewer points out, this second property is critical for learning on systems of biological relevance using data acquirable by typical research groups, and the current authors are not aware of any other biomolecular MLCG force-fields that have been trained on this type of data.

Reviewer Point P 3.4 — I have a suggestion, which might be a bit out of scope but makes the work stronger: How about other pretraining techniques such as mask pretraining or bond/atom deletion augmentation? It would be interesting to explore those pretraining strategies.

Author reply: We appreciate these suggestions. Pretraining strategies such as mask pretraining and bond/atom deletion augmentations have been reported for models created for predicting properties of molecules as well as those generating molecules directly from scratch for given tasks. Masking is also vital for training transformer models (e.g., large language models). However, the mentioned pretraining approaches are not directly relevant to MLCG models with CGSchNet, where every CG bead needs to be present in order to define and compute the multibody potential of mean force at a given resolution. Due to the multibody nature of the prediction target, it is not immediately obvious to us what labels (e.g., energy/forces) we should expect the model to predict in the pretraining stage. In other words, the current setup does not allow us to remove some part of the molecule for augmentations.

However, during our new application to Ubiquitin, we developed a regularization strategy for training MLCG models. In brief, for each mini batch of training, we apply the following to a small proportion of data points: an isotropic Gaussian noise (with a much higher variance than the level used for noise forces) is generated and added to the coordinates, and the force label to match is set to zero. This regularization discourages the neural network from predicting high-magnitude forces outside of the known data distribution, which stabilizes the models during long CG simulations by preventing sudden blow-ups due to very wrong force predictions. We have now noted this procedure in the Supplementary Information (Supplementary Information lines 194-197).

Reviewer Point P 3.5 — In equation 4 and for total loss, I would like to see a regularization factor (Λ) to see the effect of MD force versus noise.

Author reply: There is indeed a balance to strike between the information extracted from MD forces and that from noise; this balance produces a trade-off between information derived from the true underlying Boltzmann distribution and that from the observed distribution of training samples. There are two hyperparameters to set that affect this balance: one is the variance of the noise (noise level) embedded in the definition of κ and the other (the “ Λ ”) is implicitly included in the extended force map T in equation 3. For each noise level, the corresponding regularization (Λ) is set by using an optimized form of T , to minimize the force-matching noise as we have previously proposed (see J. Chem. Phys. Lett., 14:17, 3970-3979 (2023)). Essentially, we bind these two choices together: The selection of the noise level, along with the given training dataset, determines T . In the manuscript, we have explored the effect of modifying the noise level, and have additionally compared this to the case of a non-optimized noise-only T (the noise-only cases) and the force-only case. As a result, we believe that we have explored the trade-offs observed by changing the balance of noise and force information.

In summary, as the balance of noise and force information is already implicit in the T projection in equation (3), there should be no additional factor in front of either term in Eq. 4 to represent this balance. To avoid this confusion, we have reworded the derivations. We also included pointers to the Supplementary Information and Fig. S1 to help clarify the theoretical meaning of these terms (Supplementary Information lines 222-232).

Reviewer Point P 3.6 — One minor comment: some of the figures in appendix were not compiled properly and are not showing up

Author reply: Thank you for pointing this out. It should be fixed now.

Response to reviewer 1

In the revised manuscript, the authors' modifications and discussions of relevant details are commendable. The newly incorporated Chignolin system and free energy surface analysis based on Rg/RMSD enhance the comparability of results. Supplementary Figures S7, S9, and S12 in the SI indeed clarify the physical significance.

However, in addressing the core issue I mentioned, that is, the comparison with the CCG method (arXiv:2305.01243v1). There are clear shortcomings, suggesting an attempt to sidestep the crucial points.

Reviewer Point P 1.1 — I explicitly requested a comparison of the loss functions and conceptual differences between the two approaches, yet the authors merely emphasised their avoidance of generative architectures (such as normalizing flows or GANs) without providing a detailed analysis of the fundamental mathematical distinctions in the loss functions. This evasion undermines the article's innovation: merely transplanting denoising score matching without elucidating its unique theoretical contribution to coarse-grained modelling significantly diminishes the methodological value.

Author reply: We apologize for not presenting a more detailed comparison before. We have now made the differences between the two methods more explicit and we have added the following part to the main text (page 12, lines 334-355, highlighted in red):

"The addition of configurational noise may impact other tasks central to CG MD. For example, substantial work has recently focused on reintroducing atomistic details into CG protein configurations using ML-based techniques [66-74]. Many approaches produce atomistic configurations that unambiguously map to their CG counterparts, consistent with traditional bottom-up CG; however, this same reconstruction task naturally incorporates additional flexibility between a given atomistic and CG configuration when interpreted with the proposed probabilistic map. Although this formulation has previously appeared as a practical relaxation for numerical optimization (e.g., Zhang et al. [69], Pang et al. [70]), it can instead be viewed as an exact solution in this probabilistic setting. This compatibility between reconstruction and the proposed resolution coupling may in turn improve methods which directly utilize said reconstruction to accelerate conformational sampling. In this context, recently, Zhang et al. [69] proposed learning the fine-coarse resolution coupling from data, and using the learned transformation in a generative fashion for efficient sampling in an active learning scheme. However, we note that the full application of these methods to explicit solvent molecular systems often requires complete reconstruction of the all-atom resolution, including the placement of water molecules for each protein conformation according to the Boltzmann distribution. To our knowledge, such a reconstruction is still absent from typical procedures. In contrast, the focus of the present work is on the efficient training of machine-learned coarse grained models in a data-sparse regime without reference to reconstruction; the learned effective energy function for the coarse-grained system can then be used with physics-based methods (e.g., MD) to sample the conformational landscape and estimate free energies differences."

We also included an additional document for review, in which we make a very detailed comparison between the two methods in terms of conceptual, mathematical, and application points of view. We believe that such a detailed comparison is too specific for the main text or the Supplementary Information of this manuscript.

Reviewer Point P 1.2 — While the authors cite an article (Nat. Chem. 2025, 7, 1284-1292) as evidence of “extrapolating”, they fail to present comparable data in this paper.

Author reply: While the training of a sequence-extrapolating machine-learned transferable model similar to the one of the cited manuscript could be a valuable application, we consider it to be out of scope of the current manuscript. The goal of this work is not to tackle the long-standing problem of chemical transferability in coarse grained models but to present a training methodology that significantly improves the data-efficiency of current machine-learned, coarse-grained models.

Furthermore, we believe that the Ubiquitin model presented in the revised version of our manuscript demonstrates the extrapolation capabilities of our method. The trained model is capable of sampling the free energy landscape of the protein, including the unfolded state and folding-unfolding transitions, which are not present in the training data.

Reviewer Point P 1.3 — If the core advantage of noise-assisted training is reducing sample requirements, then a direct comparison should be made between CCG and the present method on identical systems (e.g., Chignolin) to determine the minimum dataset size needed to achieve equivalent accuracy. If the authors cannot obtain the CCG code to evaluate the actual system, they should at least compare the efficiency of the two methods on a numerical toy model.

Author reply: Indeed, the code associated with the CCG preprint is not available, nor does the preprint disclose any details on the training data of their models and the supplementary information is not available on the arXiv (e.g., there is no mention of the number of datapoints used in the training). We assume they used the full training dataset from the work of Lindorff-Larsen et al. (Science 334, 517 (2011)) for their Chignolin model. In contrast, we mention in lines 314-316 of Supplementary Information that in our work we are able to create our MLCG Chignolin model with only 4% of such dataset, clearly showing the minimal sample requirement of our approach. Given the absence of a usable CCG code, a re-implementation from scratch for a more direct comparison is not a reasonable request.

Reviewer Point P 1.4 — In addition, regarding the expansion of application scenarios, the authors’ response to the rare event sampling problem is equally unconvincing. Yet the authors merely vaguely mention “can be combined with data generated by enhanced sampling” and evade specific case comparisons by citing the lack of all-atom reconstruction methods. In fact, the performance of coarse-grained models in rare events (such as sampling folding transition states) can be evaluated through the statistical properties of their own trajectories, without relying on FCR functionality.

Author reply: We thank the reviewer for their comments about the need to address rare-event sampling such as folding transition states. We have now included in the Supplementary Information a more detailed analysis regarding the sampling of transition states. We do this by showing the average fractions of native contacts $\langle Q_i \rangle$ formed in different segments of the protein at each stage of the folding process (identified by the global fraction of all native contacts formed, Q). As it can be seen in Supplementary Figure S13, our models sample the folding transition states and the folding process, forming the contacts in the same order as the atomistic reference (for the proteins for which we have an atomistic reference). We refer to these results in the main text, at page 6 (lines 191-196, highlighted in red):

“An analysis of the fraction of native contacts shows that the formation order of different structural motifs are well recovered by the models, demonstrating an application in rare-event sampling of folding-unfolding transition states (see Supplementary information IIE). We also include in the Supplementary

Information the two-dimensional free energy surfaces for models and references over TIC1–TIC2 as well as RMSD– R_g (radius of gyration)."

And also at page 9 (lines 236–237, highlighted in red):

"At the model is also capable of reproducing the majority of the native contact trends (see Supplementary Information)."

Reviewer Point P 1.5 — Therefore, the current revision has not yet met my previous modification requirements, particularly failing to clearly delineate the boundaries between this paper's methodology and CCG through quantitative comparisons and theoretical analysis. If these analyses cannot be completed, the assertion of "innovation" should be toned down, explicitly positioning this work as a specific application of denoising score matching within coarse-grained force fields.

Author reply: We understand the comments from the reviewer regarding the claims about the novelty of our work. We consider that with our current reply to the comments and the modifications to the latest version of our manuscript, the differences to the CCG method are shown clearly, and we believe the tone of our claims is appropriate.

Response to reviewer 1

Reviewer 1 asked us for a more detailed comparison with our approach and the CCG method. We include here a very detailed comparison between the two methods in terms of conceptual, mathematical, and application points of view. We believe that such a detailed comparison is too specific for the main text or the Supplementary Information of our manuscript, as the methods are quite different in scope and applicability.

1 Conceptual differences

The Cycle Coarse Graining (CCG) [1] proposes to simultaneously solve the “fine-grained reconstruction” (FGR) problem to obtain a fine-grain/high resolution (FG) configuration from a coarse-grained (CG) one along with the “sampling problem” of sampling configurations from complex free energy landscapes. The authors model the conditional probability $p(\mathbf{r}|\mathbf{R})$ of having a FG configuration \mathbf{r} given a CG configuration \mathbf{R} through a conditional generative model. Meanwhile, our methodology proposes a technique to combine forces arising from a noising procedure and forces at the FG resolution to learn an effective energy function at CG resolution, which can be then used for free energy estimation and rare event sampling. While both methods target the CG free energy surface, the CCG method estimates the free energy via a “variational free energy” estimator and requires direct access to the exact potential of the reconstructed fine grained configuration. Such a potential is only computable when the FGR is used to reconstruct the all-atom resolution. For an explicit solvent system, this means the solvent environment also needs to be reconstructed for each solute conformation, which is an ongoing research topic on its own and thus limits the applicability to realistic biomolecular systems. For instance, the CCG manuscript does not include a free energy surface estimation for the Chignolin example. On the contrary, our method directly learns the CG free energy without that restriction.

The conditional generative model learned in the CCG approach is trained using a loss function given by a linear combination of two Kullback-Leibler divergences: an energy-based term (common in the context of flow-based generative models), which requires an energy estimator for the FG system, and a data-driven term, which only requires a good sample of the FG distribution.

After machine learning a model for $p(\mathbf{r}|\mathbf{R})$, the authors use this conditional probability in two different application scenarios:

1. If the distribution $p(\mathbf{R})$ of the coarse grained variable is also known, the distribution of the fine grain $p(\mathbf{r})$ can be sampled by factorized sampling: first sample \mathbf{R} following $p(\mathbf{R})$ and then sample from $p(\mathbf{r}|\mathbf{R})$. This application is unrelated to our work.
2. The free energy in the CG space $F(\mathbf{R})$ can be estimated through a “variational free energy” estimator $F_\theta(\mathbf{R})$. This application is more aligned with our work.

The authors apply CCG to three systems, one of which could be compared to our study: A cascade model of protein Chignolin at two coarse resolutions, one mapping from its atomistic configuration to a C_α configuration, and a second one mapping the C_α configuration into the RSMD to the native state. For this Chignolin example, neither of the two application scenarios above were fully explored. The FGR does not reproduce the full all-atom resolution (protein and solvent) of the original model, and the application is limited to the reconstruction of protein-only configurations from coarse-grained configurations sampled from the training data. A comparison of our approach with this application is impossible as they are orthogonal: the CCG approach gives the (partial) FGR but does not produce a CG free energy, while we obtain a CG free energy but can not do FGR. The authors did not attempt to obtain an estimate of $p(\mathbf{R})$, and they used the distribution from their own training data as the estimation instead. This is fundamentally different from our methodology, in which we estimate the free energy directly by sampling via molecular dynamics simulations from the learned effective CG energy function. The authors provided no discussion or treatment of the solvent molecules in the system, which are part of their training data [2] composed by long-time, explicit-solvent molecular dynamics simulations.

2 Loss function

2.1 General Loss

Following equations (5), (6) and (S5) of the manuscript [1], the general loss function for a CCG model is expressed as:

$$L(\theta) = (1 - \eta_1)D_{\text{KL}}(q_\theta \parallel p) + \eta_1 D_{\text{KL}}(p \parallel q_\theta), \quad (\text{C1})$$

with q_θ the probability distribution based on mapping latent variable z bijectively to the fine-grained variable \mathbf{r} , p the ground truth conditional distribution of \mathbf{r} and η_1 a parameter that modulates the contribution

of each loss term. Taking the gradient with respect to the parameters, we can express the equation more explicitly as:

$$\nabla_{\theta} L(\theta) = (1 - \eta_1) \mathbb{E}_{\mathbf{r} \in \mathcal{D}} [-\nabla_{\theta} \log q_{\theta}(\mathbf{r}; \mathbf{R})] \quad (\text{C2})$$

$$+ \eta_1 \mathbb{E}_{\mathbf{r} \sim q_{\theta}} [\nabla_{\theta} \log q_{\theta}(\mathbf{r}; \mathbf{R}) + \nabla_{\theta} U(\mathbf{r}; \mathbf{R})], \quad (\text{C3})$$

with U being an ‘‘oracle’’ energy estimator or energy label. Note that we assume that $\beta = 1$ for the derivation for the sake of simplicity. The authors refer to equation (C3) as the *energy based objective* and to (C2) as the *data-driven objective*.

While term C2 presents some similarity with our loss function as expressed in equations (3) and (4) of our manuscript (i.e. $\mathbf{T}_{\text{noise}}$), both losses differ fundamentally in several aspects:

1. Our work doesn’t use reference energy labels, nor their derivatives w.r.t. model parameters, for training as the relation between coarse grained energy and atomistic energy is through a high-dimensional integral, whose estimation we want to bypass due to the large number of degrees of freedom removed from our system (the solvent coordinates).
2. Instead of learning a function that estimates a transformation from CG to atomistic variables directly, we use a probabilistic atomistic to CG mapping as a tunable tool to obtain an energy function which will accurately reproduce the CG configurational distribution.
3. While the scalar parameter η_1 gives a degree of control over the mixing of the noising contributions and information in the atomistic energy function, our methodology allows for the possibility of combining both terms in a more complex manner. We optimize a linear \mathbf{T} mapping, which operates over an extended phase-space that combines both the contributions of force associated with the noising procedure (‘‘noise forces’’) and the atomistic forces projected over the CG space in a component-wise way. Furthermore, this combination happens before the expectation of norm operation, avoiding instabilities that could arise due to different magnitudes in the different loss terms.

For more information about our training loss in depth, we refer to section I of both the main text and the supplementary information. We would like to highlight that the core novelty of our methodology is the combination of the signals from both the noise contribution and the atomistic projections which, as shown in the results section, can significantly reduce the data requirements of a model and enables the crossing of very high energy barriers.

2.2 Chignolin model

The Chignolin model presented in section III.2 of [1] is trained using only the *data-driven objective*. This is consistent with using the training data from [2], which does not have energy or force data labels. Then, the gradient of the loss function takes the form:

$$\nabla_{\theta} L_{\text{CLN}}(\theta) = \mathbb{E}_{\mathbf{r} \in \mathcal{D}} [-\nabla_{\theta} \log q_{\theta}(\mathbf{r}; \mathbf{R})] \quad (\text{C4})$$

The authors make use of their so-called *consistency regularization*, which uses a gaussian kernel to approximate the Dirac delta function. The gradient of the loss function then takes the form:

$$\nabla_{\theta} L_{\text{CLN}}(\theta) = \mathbb{E}_{\mathbf{r} \in \mathcal{D}} \left[-\nabla_{\theta} \log q_{\theta}(\mathbf{r}; \mathbf{R}) + \sigma \nabla_{\theta} \|\mathbf{M}\mathbf{r} - \mathbf{R}\|^2 \right] \quad (\text{C5})$$

With σ the standard deviation of the gaussian kernel. On top of the differences discussed in the previous section, for this particular loss function, we highlight additional differences with our methodology:

1. Instead of using the *consistency regularization* to approximate the Dirac delta of the mapping procedure as a Gaussian kernel, we sample a set of conformations from the noised distribution to estimate the noise component of the forces.
2. We also use information from the atomistic forces projected into the CG space, that is absent in their training data.

3 Sampling

In the case of Chignolin, the CCG method samples a trajectory across a high-resolution space by fixing the random variable of the generative model and then generating different high-resolution structures for a selected set of CG values. Changing the value of the random variable z gives rise to a new of ‘‘trajectory’’, allowing the generation of an ensemble of ‘‘trajectories’’ that resemble physical trajectories.

On the other hand, our methodology directly produces a energy function which can be sampled according to traditional simulation methods such as Langevin dynamics, which generates time-correlated real trajectories. It also facilitates various enhanced sampling methodology such as Parallel Tempering, the samples from which can be reweighted to produce unbiased estimators of the free energy and any other associated observable.

References

- [1] Zhang, J., Lin, X., Gao, Y. Q. *et al.* Invertible coarse graining with physics-informed generative artificial intelligence. *arXiv preprint arXiv:2305.01243* (2023).
- [2] Lindorff-Larsen, K., Piana, S., Dror, R. O. & Shaw, D. E. How fast-folding proteins fold. *Science* **334**, 517–520 (2011).